

# Estimates of the aerosol indirect effect over the Baltic Sea region derived from twelve years of MODIS observations

Giulia Saponaro[1], Pekka Kolmonen[1], Larisa Sogacheva[1], Edith Rodriguez[1], Timo Virtanen[1] and Gerrit de Leeuw[1,2]

[1]Finnish Meteorological Institute, Helsinki, 00560, Finland
[2]Department of Physics, University of Helsinki, Helsinki, 00560, Finland

*Correspondence to:* Giulia Saponaro (giulia.saponaro@fmi.fi) and P. Kolmonen (pekka.kolmonen@fmi.fi)

**Abstract.** Twelve years (2003-2014) of aerosol and cloud properties retrieved from the Moderate Resolution Imaging Spectroradiometer (MODIS) on-board the Aqua satellite were used to statistically quantify aerosol-cloud interaction (ACI) over the Baltic Sea region including the relatively clean Fennoscandia and the more polluted Central-Eastern Europe. These areas allowed us to study the effects of different aerosol types and concentrations on macro- and microphysical properties of clouds: cloud effective radius (CER), cloud fraction (CF), cloud optical thickness (COT), cloud liquid water path (LWP) and cloud top height (CTH). Aerosol properties used are aerosol optical depth (AOD), Ångström Exponent (AE) and aerosol index (AI). The study was limited to low level water clouds in the summer.
The vertical distributions of the relationships between cloud properties and aerosols show an effect of aerosols on low-level water clouds. CF, COT, LWP and CTH tend to increase with aerosol loading, indicating changes in the cloud structure, while the effective radius of cloud droplets decreases. The ACI is larger at relatively low cloud top levels, between 900 hPa and 700 hPa. Most of the studied cloud variables were unaffected by the lower tropospheric stability (LTS) except for the cloud fraction.
The spatial distribution of aerosol and cloud parameters and ACI, here defined as the change in CER as a function of aerosol concentration for a fixed liquid water path (LWP), shows positive and statistically significant ACI over the Baltic Sea and Fennoscandia, with the former having the largest values. Small negative ACI values are observed in Central-Eastern Europe, suggesting that large aerosol concentrations saturate the ACI.

**Key words:** aerosols, cloud effective radius, aerosol indirect effect, satellite

## 1 Introduction

Aerosols and especially their effect on the microphysical properties of clouds are among the key components that influence the Earth's climate. As the magnitude and sign of such effects are not well known, understanding and quantifying the influence of aerosols on cloud properties constitute a fundamental step towards understanding the mechanisms of anthropogenic climate change (IPCC, 2013).

As aerosols may act as cloud condensation nuclei (CCN), an increase in their number concentration can lead to an increase in the number of cloud droplets in super saturation conditions and a decrease of the cloud droplet radius. The decrease of the droplet effective radius resulting in an increase of the cloud albedo, under the assumption of a constant liquid water path, is known as the Twomey effect (Twomey, 1977). The decrease of droplet size can also impact the precipitation cycle, as the smaller droplets require longer time to grow into precipitating droplet sizes. Additionally, a possible decrease of the precipitation frequency of liquid clouds increases the lifetime of clouds (Albrecht, 1989). These impacts of aerosols are called the first and second indirect effects, respectively.

A quantitative evaluation of the effects of aerosols on clouds may be possible mainly in a statistical sense because of the local interactions between meteorological conditions and aerosols (Tao et al., 2012). Satellite-based remote sensing instruments can provide a large data set for statistical analysis from long-term observations of the aerosol indirect effect on a large spatial scale with daily global coverage, complementing localized ground measurements and providing necessary parameters for climate models.



A common approach in the satellite-based investigation of the first indirect effect is the concept of the aerosol-cloud-
interaction (ACI) that relates the cloud optical thickness (COT), cloud effective radius (CER) or cloud droplet number
concentration (CDNC) to the aerosol loading. The aerosol loading is usually expressed by the aerosol optical depth
(AOD) or aerosol index (AI, defined in Section 3) that are used as a proxy for the CCN concentration.
Many studies describe the interaction between aerosols and clouds through the correlation of the satellite retrieved
aerosol concentration and cloud droplet size on a global or regional scale. Inverse correlations on a global (Breon et al.,
2002; Myhre et al., 2007; Nakajima et al., 2001) and a regional scale (Costantino et al., 2010; Ou et al., 2013) have been
found while Sekiguchi et al. (2003) and Grandey and Stier (2010), applying satellite data on a global scale, found either
positive, negative, or negligible correlations between the CER and AOD depending on the location of the observations.
Jones et al. (2009) emphasized that the ACI should be inferred in aerosols or cloud regimes determined on a regional-
scale, as the relevance of aerosol type, aerosol concentration, and meteorological conditions differ around the world.
Areas located at high latitudes are excluded from most of the studies due to a seasonal limitation of the satellite
coverage and a smaller number of observations when compared to the global averages over the year. Lihavainen et al.
(2010) compared in-situ and satellite measurements to quantify the aerosol indirect effect on low-level clouds over
Pallas (Finland), a northern high-latitude site, and concluded that the ACI values derived from ground based
measurements were higher than those obtained from satellite observations. Unlike the in situ instruments, the
wavelengths used in the satellite retrievals constrain the detection of fine particles to those larger than about 100 nm,
thus making it impossible to account for all CCN. Sporre et al. (2014a, 2014b) combined aerosol measurements from
two clean, northern high-latitude sites with satellite cloud retrievals and observed that the aerosol number concentration
affects the CER while no impact on the COT was observed. As both studies focused on specific locations, no
information was thus provided on a larger scale in the Baltic region. This work investigates whether the first indirect
effect can be observed also by means of satellite-derived observations over the region of Baltic Sea Countries, a region
that offers a northern clean atmospheric background (Fennoscandia) contrasted by a more polluted one (Central-Eastern
Europe).
Twelve years of aerosol and cloud properties available from the Moderate Resolution Imaging Spectroradiometer
(MODIS) retrievals were investigated on a regional scale to determine whether it is possible to observe the response of
the properties of low-level liquid clouds to different aerosol loadings in different atmospheric conditions.
The satellite retrieval products are introduced in Sect. 2, the approach adopted for the aerosol-cloud interaction analysis
is described in Sect. 3, and the results of the analyses are presented in Sect. 4.
**2 Data**
The area covered in this study is situated at high latitudes (50º N, 10º E, 70º N, 35º E). At these latitudes the solar zenith
angle (SZA) constrains the available satellite dataset: a large value of the SZA implies higher uncertainties on the
retrieved parameters. Due to the SZA and data coverage constraints, we limit the dataset to summer season (June, July,
August) observations that have been collected by the MODIS instrument between 2003 and 2014. Data are analysed
only from the MODIS/Aqua platform that crosses the equator at 13:30 local time, when the clouds are fully developed.
The MODIS Collection 06 Level 3 (C6 L3) product provides cloud and aerosol parameters at daily time resolution and
at a regular 1º x 1º degree spatial grid. The application of MODIS satellite data to aerosol-cloud interaction studies is
often criticized for the lack of coincidental aerosol and cloud retrievals. Studies such as Avey et al. (2007), Breon et al.
(2002) and Anderson et al. (2003) showed that in the case of daily products at 1º x 1º degree resolution it is unnecessary
to individually couple the aerosol and cloud measurements. Therefore, in this study aerosol and cloud data are assumed
to be co-located.
The MODIS C6 L3 product includes cloud microphysical parameters (CER, COT, LWP) with statistics (mean,
minimum, maximum, standard deviation) determined at three different wavelengths (1.6, 2.1 and 3.7 μm) for each
cloud phase (liquid, ice, undetermined) separately.
We filtered the MODIS cloud data according to the following criteria:



▪ Cloud parameters were considered only in the liquid-phase.
▪ To eliminate possible outliers, retrievals with a standard deviation higher than the mean values were
discarded.
▪ Observations with a mean cloud top temperature less than 273 K were eliminated to ensure only warm liquid
cloud regimes.
▪ The multi-layer flag was applied to select only single layer clouds.
▪ Transparent-cloudy pixels (COT < 5) were discarded to limit uncertainties (Zhang et al., 2012).
▪ The CER derived from the 3.7 μm wavelength was chosen as it has been shown to be less affected by the sub-
pixel heterogeneity (Zhang et al., 2012).
▪ To exclude precipitating cases, observations were discarded when the difference between CER at 3.7 μm and
CER at 2.1 μm was greater than 10 μm (Zhang et al., 2012).
The science data sets (SDS) for the atmospheric aerosol information in the MODIS C6 L3 provides the AOD retrieved
at several wavelengths and as a product from the application of either the 'Deep Blue' or 'Dark Target' algorithm, or a
combination of both retrievals (Levy et al., 2013; Sayer et al., 2014). The SDS
'Aerosol_Optical_Depth_Land_Ocean_Mean' is the solely product providing the AOD at 0.55 μm globally, while the
other aerosol SDSs provide the AOD over land and water separately. As C6 provides the Ångström Exponent (AE) over
land only, the AOD at the wavelengths of 0.46 and 0.66 μm present in both 'Aerosol_Optical_Depth_Land_Mean' and
'Aerosol_Optical_Depth_Ocean_Mean' were used to derive the AE globally as shown in Sect. 3.
To assess the effect of meteorological conditions on cloud properties the ECMWF ERA-Interim re-analysis data were
applied to derive the Lower Tropospheric Stability (LTS). Although not a ready-to-use product, the LTS is computed as
the difference between the potential temperature at 700 hPa and at the surface (Klein and Hartmann, 1993) describing
the magnitude of the inversion strength for the lower troposphere.

## 3 Methods

After selecting the cloud parameters as listed in the previous section, the number of observations were binned for both
aerosol and cloud products. From the obtained histograms, the 95 % of the most frequent ranges were selected from the
total dataset by filtering out 2.5 % of data from the extremes. These statistically more robust datasets were used in
further analysis.
The product of the AOD, representing the column-integrated optical extinction of aerosol at a given wavelength, and the
derived AE, describing the spectral dependency of the AOD, results into a third aerosol property of interest, the aerosol
index (AI). The AI is used as a proxy for the fine mode aerosol particles which have a larger contribution to the CCN
than the coarse mode particles (Nakajima et al., 2001). MODIS Collection 6 provides the AE only over land. To
homogeneously estimate the AI over the Baltic Sea and the surrounding land areas, the AE is evaluated by applying
equation:
$$AE = -\log(AOD_{\lambda_1}/AOD_{\lambda_2})/\log(\lambda_1/\lambda_2),$$
35 (1)

to the wavelength pair of $\lambda_1 = 0.66$ μm and $\lambda_2 = 0.46$ μm which are available both over land and over sea. The C6
MODIS aerosol algorithm does not, however, allow the determination of the AE for coastal and inland water regions
(Levy et al. 2013). This would leave large parts of the Baltic region under investigation in this work out of the analysis
(see Fig.2 b and c). For this reason the aerosol-cloud interaction was analysed, in addition to the AI, also with the AOD.
Seasonal mean values of aerosol (AOD, AE, AI) and cloud parameters (CER, CF, COT) were computed for the period
of 2003-2014.
Aiming to observe how the variation in aerosol conditions influences cloud properties, we adopted the approach of
Koren et al. (2005) to analyse the average vertical distribution of the relationships between aerosols and cloud





properties. The AOD and AI datasets were firstly sorted in ascending order and successively divided into five equally-
sampled classes that represent the averages of aerosol conditions for each of the classes. The cloud properties were then
divided according to these AI and AOD classes and plotted as functions of cloud top pressure.
The response of the cloud properties to clean versus polluted aerosol conditions was studied spatially. The 25[th] and 75[th]
percentiles of the AI and AOD (AI/AOD) were computed for each spatial grid point, the former constituting the upper
limit for the AI/AOD values representing low aerosol loadings and the latter the lower limit for the AI/AOD values for
heavy aerosol loadings. These percentile values were then used to choose cloud parameters for clean and polluted
aerosol conditions. The difference between a cloud parameter value in low and high aerosol conditions is:
$\Delta \mathrm{Cloud\_X} = \mathrm{Cloud\_X}_{\mathrm{25th\ percentile}} - \mathrm{Cloud\_X}_{\mathrm{75th\ percentile}},$
10                                      (2)

where the considered cloud parameters, Cloud_X, are the cloud effective radius, cloud top pressure, cloud optical
thickness, cloud fraction and liquid water path. The subscripts indicate that the cloud parameter is representative for
clean atmospheric conditions, $\mathrm{Cloud\_X}_{\mathrm{25th\ percentile}}$, or for polluted atmospheric conditions, $\mathrm{Cloud\_X}_{\mathrm{75th\ percentile}}$. The
difference of these two variables shows which aerosol condition has a larger effect on cloud properties.
Matsui et al. (2006) found that aerosols impact the CER stronger in an unstable environment (low LTS) than in a stable
environment (high LTS) where the intensity of the ACI is reduced due to the dynamical suppression of the growth of
cloud droplets. Following this result, we also compared cloud microphysical properties with both the AI/AOD and the
LTS.
The area of this study was divided into three sub-regions as presented in Fig. 1: Area 1 covers the Baltic Sea, while
Area 2 and Area 3 include only land pixels over Fennoscandia and Central-Eastern Europe, respectively. Figure 2
shows time series of the summer averages of the AOD and AI computed for each sub-region. It is easy to see in Fig. 2
that these three areas have generally different aerosol conditions: within the land sub-regions, the lower AI and AOD
averages occur over Area 2 while over Area 3 these values are higher during the entire period. Area 1, the Baltic Sea, is
considered as a third sub-region per se due to the dominance of maritime aerosol conditions.
The ACI related to the CER was computed using the formulation from McCominsky and Feingold (2008):
$\mathrm{ACI} = -\left.\dfrac{\partial \ln \mathrm{CER}}{\partial \ln \alpha}\right|_{\mathrm{LWP}},$
27    (3)

which indicates how a change in the CER depends on a change in the aerosol loading α, given by either the AI or the
AOD, for a constant LWP. The ACI was computed by dividing the CER and the AI/AOD over LWP bins ranging from
20 to 300 g m$^{-2}$ with an interval of 40 g m$^{-2}$ and then by performing a linear regression analysis with the logarithms of
the CER and α in each LWP bin. Two approaches were applied to present the ACI: in the first, the ACI were obtained
for each sub-region and plotted as a function of the LWP while in the second approach the ACI was computed in a 2º
spatial grid. In the grid approach we chose the LWP interval that provided statistically significant ACI estimates for
each of the three sub-regions. The statistical significance is determined by the null-hypothesis test scoring a p-value <
0.05 (Fischer, 1958).
**4 Results**
The time series in Fig. 2 shows the summer averages for the AOD and AI between 2003 and 2014. The AI is highest
over Area 3 (Central-Eastern Europe), with an overall AI mean value of 0.29 ± 0.03 (regional mean ± standard
deviation), followed by Area 1 (Baltic Sea), 0.20 ± 0.02, while over Area 2 (Fennoscandia) the lowest AI mean value of
0.16 ± 0.01 is found. Area 3 also presents the highest averages for the AOD, 0.22 ± 0.02, but Area 2 and Area 1 have
comparable AOD values: 0.16 ± 0.02 and 0.14 ± 0.01, respectively.
The spatial variations of the aerosol and cloud properties are shown in Fig. 3. A decreasing south-north gradient of
AOD is observed in Fig. 3a where the highest values are found over Area 3 (Northern-Germany and Poland), and over



Area 2 (the Atlantic coast of Norway). While no discontinuities can be seen for the AOD distribution over Area 1 and
Area 2, a clear distinction is evident in the AE (Fig. 3b). Indicating the dominance of fine particles, high values of the
AE are found over the entire Area 1, over the Eastern part of Area 3, and over the North-Western part of Area 2. Low
values (AE < 1) are only found over the land areas 2 and 3. The validity of the MODIS AE over land is generally
considered unrealistic. Nonetheless, in the case of dominance of fine mode aerosols the MODIS AE agrees with
AERONET (Levy et al., 2010) while disagreements occur in coarse aerosol cases (Jethva et al., 2007; Mielonen et al.,
2011). Over ocean, a good agreement between MODIS AE and AERONET is found globally but with the limitation of
AOD > 0.2 (Levy et al., 2015), a restriction that cannot be applied in our study area where the regional AOD is about
0.2. Therefore, the high values of the AE over the Norwegian Sea are rather unlikely to be correct. Nevertheless, the AE
over Area 1 (Fig. 3b) is matching the median range of 1.46-1.49 obtained from a validation study that compares the AE
retrieved by SeaWiFS and MODIS Aqua/Terra with the three AERONET stations over the Baltic Sea (Melin et al.,
2013). The AI (Fig.3c) over Area 1 is comparable to the values over Area 3, while the lowest values occur over Area 2.
The spatial distributions of the cloud properties (COT, CER, CF) are shown in Fig. 3d-f. As in the aerosol case, Area 2
presents a distinctive discontinuity between land and water pixels (Fig3 d-f). These results are confirmed in Karlsson
(2003) where Area 1 (the Baltic Sea) exhibits low cloudiness while high cloud amounts are found over the
Scandinavian mountain range (Area 2) and the Norwegian Sea. According to the first AIE, the CER (Fig. 3e) appears to
be better correlated with the AOD (Fig. 3a) rather than the AI (Fig. 3c) and the COT maxima are also in correspondence
with the AOD minima over the coast of Norway (Area 2). Over the Norwegian coast the high values of the COT and the
CF can be explained by high hygroscopicity of sea spray aerosols, which makes these particles very efficient. Another
feature of Fig. 3e is the low effective droplet radius over Area 1 (the Baltic Sea). Unlike Area 3 (Central-Eastern
Europe), Area 1 does not match with any high aerosol loading (Fig. 3a, c) when compared to the surrounding area. In
fact, the AOD over Area 1 is as low as in Area 2 (Fig. 2), even though for these land areas the CER is about 1-2 μm
larger.
Figure 4 presents the 10-year average of the cloud properties, divided into five classes of the AI (Fig. 4a-d) and AOD
(Fig. 3e-h), respectively, plotted as function of cloud top pressure.
It can be observed that the lowest values of CTP correspond to the higher classes of AI/AOD. Assuming the CTP to be
an indicator of the cloud top height, this may suggest an enhancement of the cloud vertical structure.  This result was
also found by Koren et al. (2005) where convective clouds over the North Atlantic showed a strong correlation between
the aerosol loading and the vertical development of the clouds.
Furthermore, the cloud droplet effective radius (Fig. 4a, e) has smaller values in higher AI/AOD classes. The opposite
behaviour, lower average values corresponding to the lower classes of the AI/AOD, can be seen for the COT (Fig. 4c,
g) and LWP (Figs. 4d, h) while the CF (Fig.4b, f) is not affected by either the AI or AOD. Overall, Fig. 4 reveals that
the cloud parameters are clearly affected by the AI/AOD segregation at lower levels of the CTP. For this reason, we
limit our dataset to cloudy pixels where the CTP is between 700 hPa and 900 hPa.
In Fig. 5 the CER is plotted as a function of AI for fixed values of the LWP (five intervals as above) and the CTP
(between 700 and 950 hPa, in 50 hPa bins). The highest AI in Area 1 (the Baltic Sea) is around 0.35 for the lowest
clouds (CTP 900-950 hPa) decreasing to 0.3 for the highest clouds (CTP 700-750 hPa). Over Area 2 (Fennoscandia) the
aerosol loading is not clearly connected to the cloud height, showing a constant AI average of approximately 0.25. As
expected, Area 3 has the highest average of AI out of the three sub-regions with values as high as 0.6 for the lowest
clouds and a small decrement for the highest clouds. The cloud droplet size in Area 1 (the Baltic Sea) and Area 2
(Fennoscandia) shows a strong negative correlation with the AI, while a weak correlation is observed over Area 3
(Central-Eastern Europe). Moreover, Area 1 has no results for the high LWP bins: clouds over the Baltic Sea are most
likely stratiform clouds which are characterized by a lower LWP than for convective continental clouds. Similar results
are also found when the AOD is substituted by the AI (not shown).
Applying Eq. 2 to the cloud parameters, the impact of low and high aerosol loading (ΔCloud_X) on cloud properties
(Cloud_X) is presented in Fig. 6. Resulting from a grid-based analysis, ΔCloud_X < 0 means that the observed cloud
parameter, Cloud_X, has a larger value in polluted cases (AI/AOD > 75[th] percentile) than in clean atmospheric
conditions (AI/AOD < 25[th] percentile) for that grid cell and vice versa, when ΔCloud_X has a positive value. As similar
results were obtained by applying the AOD and AI, only the results for the AOD are shown. ΔCF (Fig. 6a) presents





only positive values suggesting that the CF is always significantly larger in the polluted atmospheric conditions. The
positive values of ΔCTP (Fig. 6d) over Area 2 (Fennoscandia) and Area 3 (Central-Eastern Europe) agree with the idea
of the vertical development of clouds for higher aerosol loadings (Fig. 4) but other factors, such as surface heating,
might be also contributing to the results: the presence of stronger turbulences over land cause the clouds to rise higher
than in the presence of lower turbulence, for example, over a cooler water surface. The CER (Fig. 6c) shows a different
behaviour over land (Area 2 and Area 3) than over water (Area 1). Over land ΔCER is predominantly negative:
although small (< 2 μm), negative values of the ΔCER indicate that the CER is larger over areas with higher aerosol
loadings than over cleaner areas. This result is in contradiction with the theory of the AIEs. The presence of aerosol
appears to have little or no effect on ΔCOT (Fig. 6b) and ΔLWP (Fig. 6e).
To understand to what extent the link between aerosol and cloud parameters are actually due to aerosols, we evaluated
the variability of low-level liquid cloud properties as function of aerosol conditions (AOD/AI) and lower troposphere
stability (LTS). Figure 7 shows the cloud properties (LWP, CER, CF and COT) plotted as a function of the LTS and
AI/AOD. While the CF shows a gradient for both direction of the LTS and the AI/AOD, the other cloud variables
(LWP, CER, COT) are mainly affected by aerosols with little to no correlation to changes in the LTS.  The LWP and
CER are negatively correlated with aerosol parameters, showing a stronger response to the AOD than to the AI. Higher
AOD values correspond to a smaller CER (Fig. f) and higher CF (Fig. 7g) which is in agreement with the AIEs, except
for the LWP (Fig. 7a) that decreases as a function of the AOD. The LWP (Fig. 7e) shows a non-monotonic response by
increasing when the AOD ranges between 0.3-0.4, because at high aerosol concentrations the cloud droplets are smaller
and less likely to precipitate, and further the LWP slightly decreases. A possible explanation of a better correlation of
the LWP with the AI than with AOD might be found by looking at the LWP vertical distributions in Fig. 4 that indicate
a more distinctive separation of  the LWP for the AI-based classes than for AOD. Although in high aerosol loading the
CF increases as cloud droplets are smaller, they are less likely to precipitate, which is in accordance with the second
aerosol indirect effect. Regardless of the correlation with aerosols, the comparison between the CF averages as a
function of CTP in Fig. 4 and the corresponding results in Fig. 5 suggest that the sensitivity of the CF to the LTS
inhibits any possibility of observing the ACI for the CF.
Figure 8 illustrates the ACI estimate for the CER (Fig. 8a) and its corresponding correlation coefficient $r$ (Fig. 8b)
calculated for the three sub-regions as a function of the LWP bins for both AOD and AI. The lines are color-coded
according to the three areas as defined in Fig. 1. The ACI estimates for Area 1 (Baltic Sea) are positive and statistically
significant throughout the entire LWP range, increasing as a function of LWP from a minimum of 0.06 to a maximum
of 0.16 and with a corresponding $r$ ranging from -0.1 to -0.53. The values of the ACI for Area 2 range between 0.02 -
0.06 with fewer statistically significant points and a smaller $r$ than in Area 1. The results collected over both Area 1 and
Area 2 appear to be little effected by whether the AOD or AI is applied in the computation of the ACI. For Area 3 two
points of the ACI results are statistically significant  but with very low values for correlations ($r< 0.1$) for the first two
bins of the LWP and, unlike the other two sub-regions, they show a negative sign.  The ACI values are statistically
significant for the three sub-regions for the first two bins of LWP and when the AOD is chosen over the AI as α. With a
combination of these requirements, we derived the spatial distribution of the ACI and  $r$  which are shown in Fig. 9.
Positive correlations are found predominantly over Area 3, and scattered over Area 2, while negative values are
covering the majority of Area 1 and, more sparsely, Area 2.
**5 Conclusions**
In this work we have studied the applicability of satellite-based information for quantifying the aerosol-cloud
interaction over the Baltic Sea region. Distinct sub-regional differences were found in the estimates of the ACI related
to the effective radius of cloud droplets. No clear ACI results were observed for the other cloud parameters which
suggest that these may be influenced by other factors, such as the local meteorological conditions. The meteorological
conditions are represented here by the LTS which was compared to the cloud parameters. The LTS is correlated with
the CF while no effect was observed upon the other cloud parameters. In particular, there is no clear evidence of the
effect of LTS on the interaction between aerosols and cloud effective radius.
One of the key aspects of this study was to find out whether a rigorously filtered Level 3 MODIS dataset can be applied
for aerosol-cloud interaction studies at a regional level. As the northerly location of the region of interest here restrains



the availability of the MODIS observations to the summer months (JJA), one of the challenges is the limited data coverage. Moreover, the selection of specific cloud regimes and the co-location of aerosol and cloud observations are additional essential key factors in building-up a robust dataset which however further decreases the amount of data-points available. As far as known to the authors, no previous results on ACI from a satellite perspective are provided over this area.

This study shows that the different aerosol conditions characterizing the Baltic Sea countries have an impact on the ACI and this can be also observed on a regional scale. According to ACI theory, polluted atmospheric conditions are connected with clouds characterized by lower cloud top pressure, larger coverage and optical thickness. However, the cloud effective radius strictly follows the AIE's theory only over Area 1 (the Baltic Sea) which agrees also with the results presented by Feingold (1997). As reported in this study, the CER retrieved in clean clouds is mainly affected by the LWP and aerosol presence while when detected under polluted conditions it additionally shows a high dependence on other factors.

The cleaner atmosphere characterizing Area 1 (the Baltic Sea) and Area 2 (Fennoscandia) reveals statistically significant and positive ACI estimates between the CER and AOD that are in agreement with the values obtained from ground-based measurements collected at the sites of Pallas and Hyytiälä in Finland, and Vavihill in Sweden (Lihavainen et al., 2010; Sporre et al., 2014b) while over the more polluted Area 3 (Central-Eastern Europe) the sensitivity to determine the ACI locally is smaller. It can be assumed that more aerosols leads to a high concentration of the CCNs and this lowers the average droplet radius as can be seen in Fig. 3e when the radius is compared between areas located South (high aerosol load) and North (low aerosol load) of the Baltic Sea.

Our analysis of the ACI for the CER shown in Fig. 8 leads to the following conclusions:

- The lowest values of the ACI can be seen over Area 3. This is also the sub-region with the highest average AOD values leading to the smallest cloud droplet size. A further addition of aerosol particles and thus possibly also CCNs does not decrease the cloud droplet size any further. Most of the ACI values are actually negative but very close to zero.

- The positive ACI values for Area 2 shows that the addition of aerosols to a relatively clean atmosphere does decrease the droplet size.

- The AI over the land areas in the study should be considered unrealistic because the average inland AE can have values below 1.

- The average AE over Area 1 has values as high as 1.4 to 1.5. These values, however, can be trusted and have been evaluated by Melin et al. (2013).

- The low CER over Area 1 requires further explanation. The most probable cause for the low values, based on the MODIS cloud retrieval, is the relatively low cloud top height over the sea. As cloud droplets generally grow in size from the cloud base towards the cloud top (McFiggans et al., 2006), Fig. 4 confirms that the average CER increases with the decreasing CTP. Furthermore, in Fig. 5 there is a distinctive lack of results for high LWP values indicating that there are fewer clouds at higher top heights. These reasons altogether lead to low values of the CER over Area 1 as the MODIS instrument retrieves the droplet radius at cloud top, and the top height CER results are low when compared to the surrounding over-land values.

- The ACI over Area 1 has considerably higher values than over the land sub-regions, and there is a difference in the magnitude between the ACI values determined using the AOD or AI. The clean maritime atmospheric conditions lead to the high sensitivity of droplet size to changes in fine particle concentrations. The AOD and AI difference in ACI, the latter being the higher, indicates that the ACI is caused by fine particles as expected.





Another way to assess the aerosol induced changes in cloud parameters would be to analyse time series to find out
whether dynamically decreasing or increasing aerosol loading has an effect on clouds. This sort of approach was not
attempted in this work.
Another important result of this work is the comparison of the ACIs obtained using the AI and AOD, chosen as proxies
for the CCN, in order to determine which option leads to more realistic results. Even though theoretically the AI would
be a better parameter than AOD to indicate the presence of fine mode aerosol particles, the impact of uncertainties of
the derived AI might be substantial.
**Data availability**
All data used in this study are publicly available. The satellite data from the MODIS instrument used in this study were
obtained from http://ladsweb.nascom.nasa.gov/index.html. The ECMWF ERA-Interim data were collected from the
ECMWF data server http://apps.ecmwf.int/dataset/data/interim_full_daily/ .
**Acknowledgements**
This research was founded by the Maj and Tor Nessling Foundation (grant no. 201600287). The authors also
acknowledge the Academy of Finland Centre of Excellence (grant no. 272041).

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





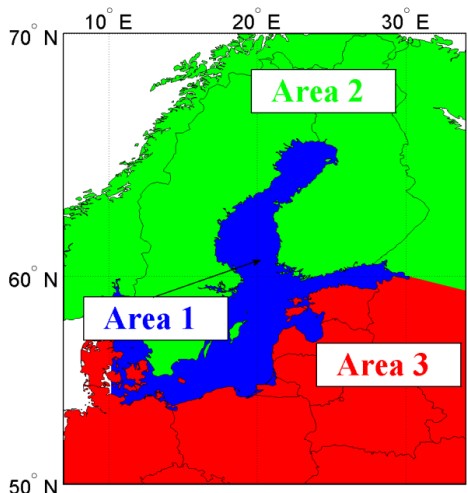

2  **Figure 1: The area covered in this study and its division into three sub-regions: Area 1, the Baltic Sea is**
3  **represented by the colour Blue, Area 2, covering the land areas over Fennoscandia, is represented by colour**
4  **Green and Area 3, in Red, includes the land areas of Central-Eastern Europe.**





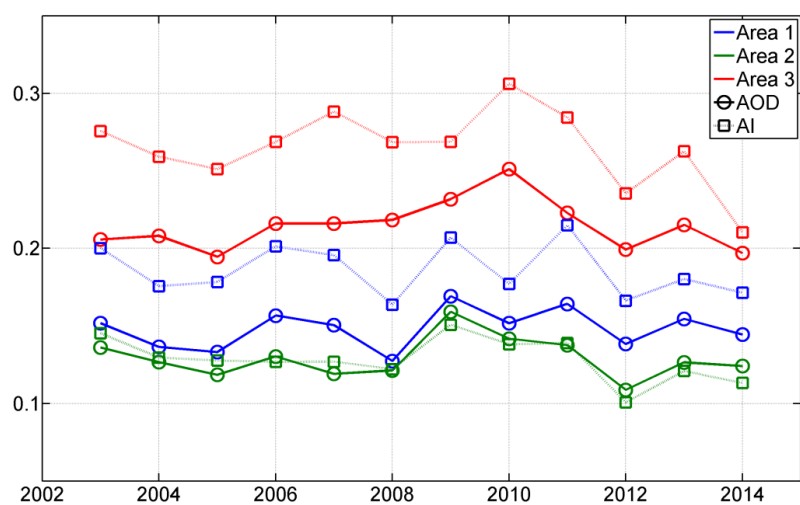

**Figure 2: Time series of summer (JJA) averages for AOD (circles) and AI (squares) for the three sub-regions. The three sub-regions are color-coded following that in Fig.1.**

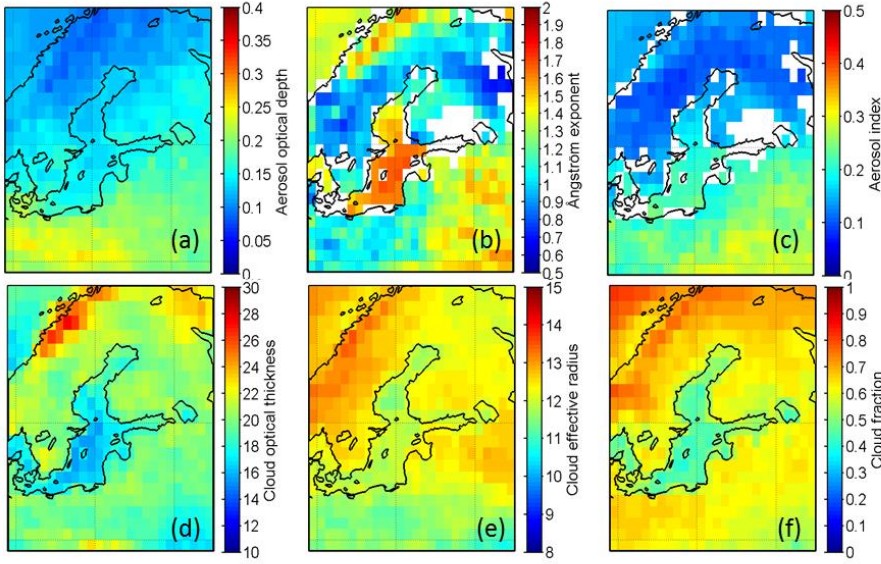

**Figure 3: Spatial distributions of AOD (a), AE (b), AI (c), COT (d), CER (e) and CF (f) averages for summer seasons between 2003-2014.**



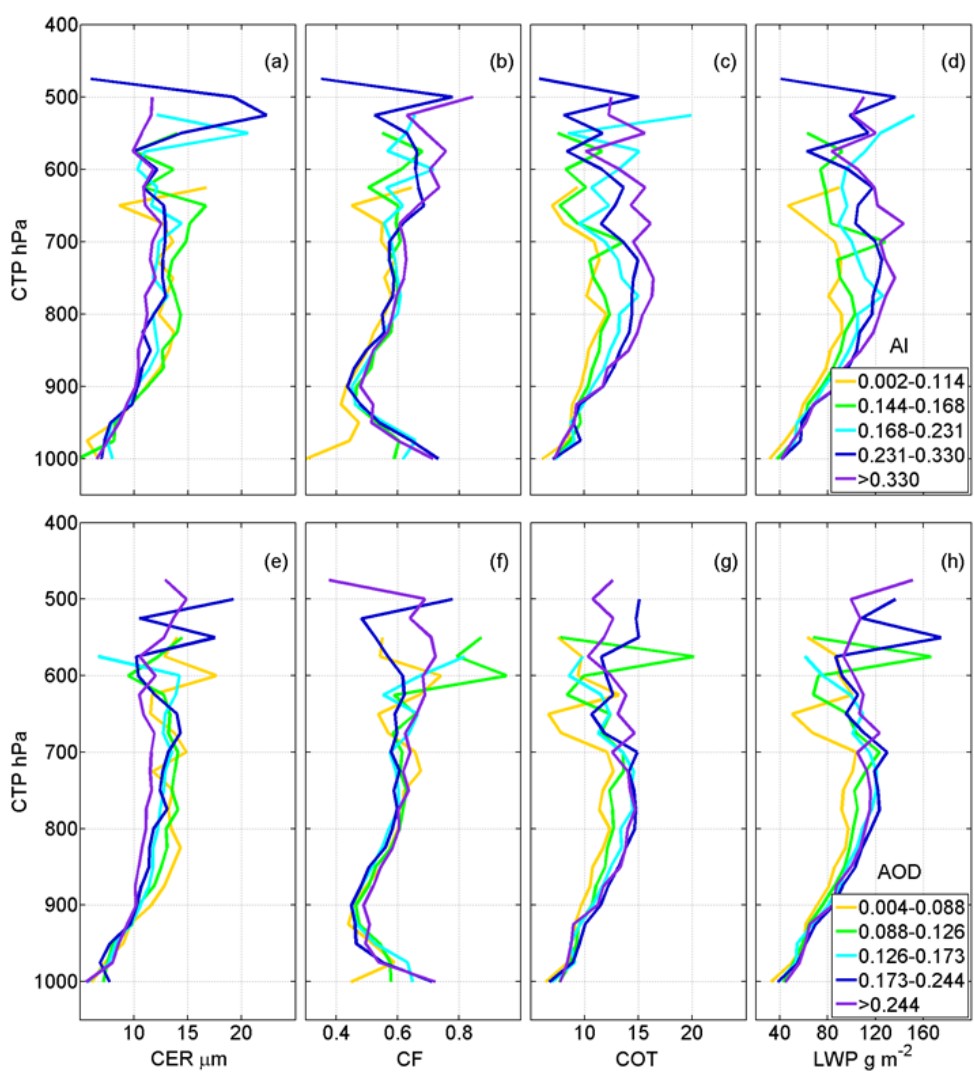

**Figure 4: 10-year averaged cloud properties as function of cloud top pressure: CER (a, e), CF (b, f), COT (c, g),**
**LWP (d, h), as functions of cloud top pressure (CTP) for five classes of AI (a-d) and AOD (e-h). Each class of**
**AI/AOD contains an equal number of samples in that interval.**





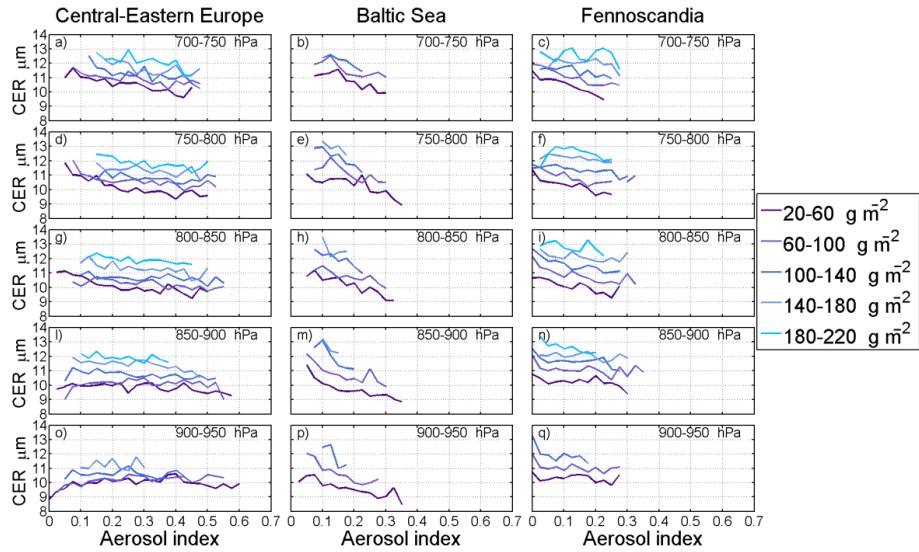

Figure 5: CER b as function of AI, stratified for subranges of CTP and LWP, for the three sub-regions. The legend on the right of the figure lists the LWP bins.

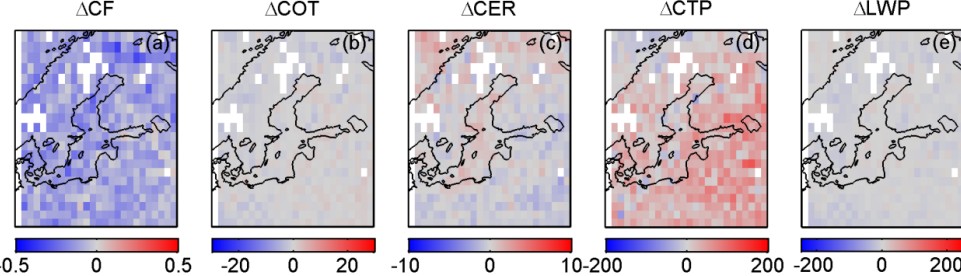

Figure 6: Spatial distributions of the difference of the cloud properties CF (a), COT (b), CER (c), CTP (d), and LWP (e) for low aerosol loading (AOD < 25th percentile) and heavy aerosol loading (AOD > 75th percentile) calculated from Eq. 2.





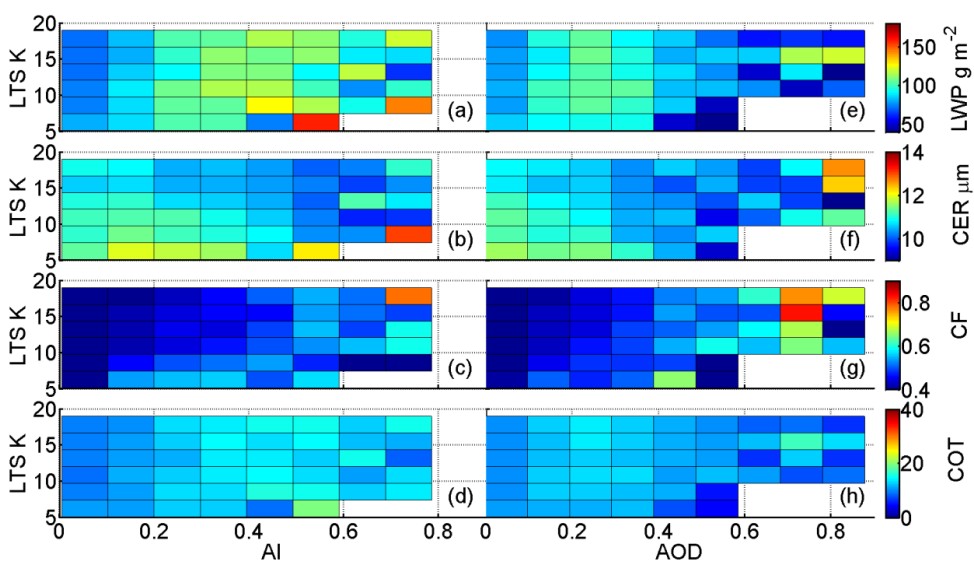

**Figure 7: Mean low-level liquid cloud properties plotted as a function of LTS and AI (a-d) or AOD (e-h).**

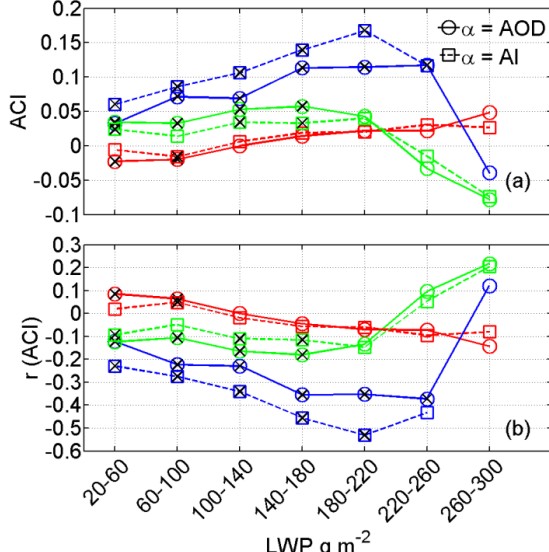

**Figure 8: ACI estimates computed for the CER as a function of the LWP and by applying both the AI and AOD**
**as proxies for the CCN are shown in (a). The correlation coefficients are presented in (b). The color-coded lines**
**refer to the three sub-regions determined in Fig.1: Area 1 (blue), Area 2 (green) and Area 3 (red) 1. The line**
**styles define whether the AOD or AI were used as the CCN proxy, α. Markers signed with a cross represent**
**points fulfilling the null-hypothesis (p-value < 0.05), hence statistically significant.**



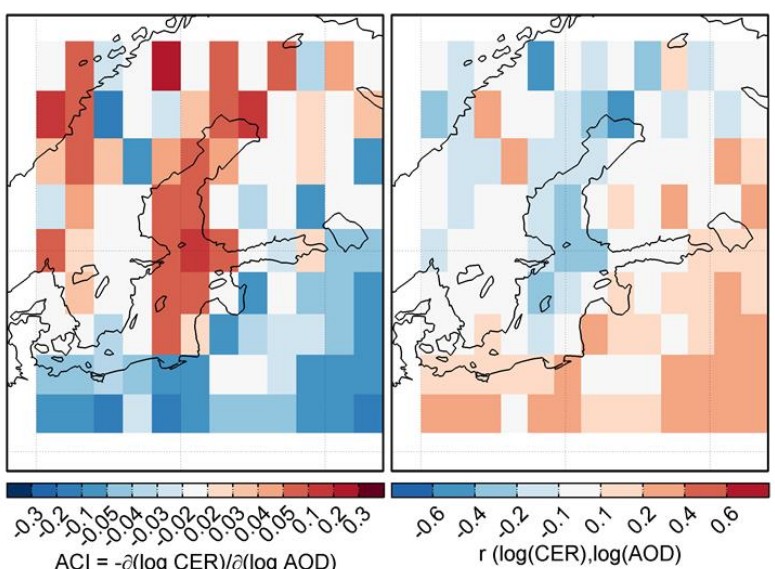

**Figure 9: Applying the AOD as a proxy for the CCN, estimates of the ACI and correlation coefficient for the CER and for the interval of the LWP between 20-60 g/m$^2$ were calculated on a grid basis. The obtained spatial distribution of the ACI is shown on the left and the correlation coefficient on the right.**

