# Peer review of "Estimates of the aerosol indirect effect over the Baltic Sea region"

_Atmospheric Chemistry and Physics, 2016_

## Referee Comment (RC1) · Anonymous Referee #1 · 3 Nov 2016

The manuscript by Saponaro et al. presents a regional satellite study on aerosol-cloud interactions over and around the Baltic Sea. This type of regional study has not been performed in this region before and the study provides new insights into aerosol-cloud interactions here. The aerosol-cloud interactions are thoroughly investigated in several different manners which highlights that different results regarding the interactions can be found depending on how the data in investigated. I suggest the manuscript is published in ACP after some minor corrections, see below.

General comments:

1. The results in the paper are somewhat inconsistent. In Fig 3 and 7 the aerosols can be seen to affect the COT and LWP while in Fig 6 no effects from aerosols are

found on these parameters. No effect on CF by the aerosols are found in Fig 3 while in Fig 6 and 7 CF is found to vary with aerosol loading. I believe the paper would benefit from a more structured discussion with regards to why the aerosol effects for different parameters appear in some of the figures while not in others.

2. Figure 3 b and d. The values of AE and COT are very high over the North western Norway. Could snow cover possibly affect the retrievals leading to high biases?

3. Figure 7: This figure is very nice and informative. Could you please change the colorbar for the COT? The colorbar goes up to 40 but the highest value in the figure is around 20. If you changed this it would be easier to see the the trends in the COT.

Technical corrections:

Page 2, line 16: 'in situ' should be changed to 'in-situ'.

Page 3, line 39: There is no figure 2 b and c.

Page 4, line 14: The sentence is somewhat awkward. Please rewrite.

Page 4, line 20-24: It seems to me that these sentences presents results and perhaps should be moved to section 4.

Page 4, line 43 – Page 5 line 1. The end of this sentence is confusing since there are no high AOD values over the Atlantic coast of Norway in figure 3a.

Page 5, line 9: "rather unlikely to be correct" awkward, please rephrase.

Page 5, line 16-17: The acronym AIE has not been defined. Also the start of the sentence is confusing, the AIE does not say that CER appears to be better correlated with the AOD.

Page 5, line 19: 'which makes these particles very effeicient' please add CCN and change the spelling to efficient.

Page 6, line 4: 'turbulences' change to 'turbulence'.

Page 6, line 10: There may be other parameters than LTS and aerosol that affects the clouds. I therefore recommend rewriting the first part of this sentence.

Page 6, line 14: To me it looks like also the CER is affected by the LTS.

Page 6, line 21-25: This part is confusing to me. There are no results on CF in figure 5 and the structures of the sentences are confusing. Could you please rewrite these sentences to clarify the reasoning with regards to CF results.

Page 7, line 7: Should ACI be AIE?

Page 7, the conclusion contains quite a bit of discussion of the results. Maybe this section should be renamed Discussion and Conclusion.

---

## Referee Comment (RC2) · Anonymous Referee #2 · 23 Nov 2016

This manuscript estimated the aerosol indirect effect over the Baltic Sea region by using MODIS L3 dataset. Over high latitude regions, such studies are very limited previously because the available dataset are often unreliable. By making use of twelve years of aerosol and cloud properties from MODIS product, the authors investigated the response of the cloud properties to change of aerosol loading based on statistical analysis, and presented some interesting findings over the region. Overall, this manuscript is well written and useful to improve our understanding on aerosol-cloud interaction. The disadvantage is lacking of the detailed explanations and discussions on the results presented (see my specific comments below).

Specific comments:

P2, Line 26-27: you raised a question here, but we don't see a clear answer finally.

P4, line 19-24, Fig.2: Area 3 AOD is much larger than AI, why?

P5, line 2-3: "Indicating the dominance of fine particles, high values of the AE are found over the entire Area 1, …', Area 1 should be dominated by sea salt, why the fine particles dominate here?

Line 18-19: 'Over the Norwegian coast the high values of the COT and the 19 CF can be explained by high hygroscopicity of sea spray aerosols,

which makes these particles very efficient'. It seems true, but why we don't see the same thing over the coast of Area 1?

Line 25, should be Fig.4e-h.

Line 26-27: Does MODIS provide cloud top height directly?

Line 32: why the CF is not affected by aerosol? any explanations?

Line 40-42: "The cloud droplet size in Area 1 (the Baltic Sea) and Area 2 (Fennoscandia) shows a strong negative correlation with the AI, while a weak correlation is observed over Area 3 (Central-Eastern Europe)", this is contradictory to our understanding.

Line 42-43: 'Area 1 has no results for the high LWP bins: clouds over the Baltic Sea are most likely stratiform clouds which are characterized by a lower LWP than for convective continental clouds', any references to present that stratiform clouds hold a lower LWP than convective clouds?

P5, line 49-p6, line 1: '$\Delta$CF (Fig. 6a) presents only positive values suggesting that the CF is always significantly larger in the polluted atmospheric conditions'. $\Delta$CF is always negative as I can see.

Line 1-3:' The positive values of $\Delta$CTP (Fig. 6d) over Area 2 (Fennoscandia) and Area 3 (Central-Eastern Europe) agree with the idea of the vertical development of clouds for higher aerosol loadings (Fig. 4)'. Higher aerosol loadings cause the vertical development of clouds, and then $\Delta$CTP should be negative, correct?

Line 6-8: 'Over land $\Delta$CER is predominantly negative: although small ($<$

2 µm), negative values of the ΔCER indicate that the CER is larger over areas with higher aerosol 8 loadings than over cleaner areas. This result is in contradiction with the theory of the AIEs", is there any explanations for this? From Fig. 3, it seems that higher CER correspond to lower aerosol loading, why the contradictory result is shown in Fig. 6?

Line 15-16: 'The LWP and CER are negatively correlated with aerosol parameters, showing a stronger response to the AOD than to the AI', CER is negatively correlated with aerosol, but LWP is NOT negatively correlated with aerosol from Fig. 7a.

Line 29-30: '…0.06 to a maximum of 0.16…', what is unit? Please keep consistent with the figure. The ACI values for Area 1 are positive, indicating a positive correlation of CER and aerosol loading, right? but why the correlation coefficients are negative?

Line 37-38: does this result means that high aerosol loading correspond to larger cloud effective radius for Area 3? Can you give some explanations?

---

## Author Comment (AC1) · 31 Jan 2017

**Response to comments from Anonymous Referee #1**
This comment addresses the comments of Anonymous Referee #1. We wish to thank the Referee for the interest in our work and the valuable inputs on the manuscript. The follow document is a point by point response in which we intend to show how we had addressed each item mentioned in the review.

Note: the following fonts are applied to divide Referee's comments from the Author's response:
*Comments from the Referee*
Response from the Authors
The page and lines numbers refer to the original version of the manuscript. The manuscript following the Author's response is the final revised version.

**Response to the general comments**
*The results in the paper are somewhat inconsistent. In Fig 3 and 7 the aerosols can be seen to affect the COT and LWP while in Fig 6 no effects from aerosols are found on these parameters.*
The colorbars of Fig.6b and e have been modified. By decreasing the lower and upper limit of the interval range, this change in color scaling allows results to be more easily visualized. Now it is possible to observe the effect of low and high AOD cases on both COT (Fig.6b) and LWP (Fig.6e). Overall, both parameters show a rather small and negligible signals. However, in details, the LWP has a predominance of (small) negative values while the COT show negative values over the majority of Area 2 and Area 3 but mixed (negative and positive values) are found over Area 1.

*No effect on CF by the aerosols are found in Fig 3 while in Fig 6 and 7 CF is found to vary with aerosol loading.*
The author misguided the Referee by stating that no aerosols effect was observed on CF in Fig.3. The Author would rather say that the signal is not very distinct because the CF lines for the aerosol classes are more 'tangled-up' compared to the profiles of the other cloud parameters.
Anyhow, Figure 3 aims, firstly, to answer the question whether aerosols have an impact on cloud vertical development. Results shows that the highest the aerosols, the lowest is the cloud top pressure (hence higher cloud tops). This effect is observable in each cloud parameter (CF, CER, COT, LWP).The effect of aerosols on CF is not missing from Fig.3, as higher aerosol loading leads to higher vertical development, but this is not a result that is directly linked, and observable, in Fig.6 and Fig.7.
Additionally, Fig. 3 also enables the reader to assess the effect of different aerosol loadings on the cloud parameters. While these are clearly visible for CER, COT, LWP, the signal is not as clear and distinctive for CF but is not absent either. The CF for the highest AOD (purple line) is dominantly the highest CF value throughout the vertical profile, in accordance with the AIE's theory. This results is also found in Fig. 6a and Fig. 7 a,g where high aerosol condition corresponds higher CF.
The text describing the CF results has been modified following what has been stated above.

*I believe the paper would benefit from a more structured discussion with regards to why the aerosol effects for different parameters appear in some of the figures while not in others.*
By addressing the Referee's comments, the Author hopes that the structure of the results and discussion is now improved and better articulated.

*Figure 3 b and d. The values of AE and COT are very high over the North western Norway. Could snow cover possibly affect the retrievals leading to high biases?*
Studies over both the Baltic Sea (Melin at al., 2013) and the Norwegian coastline (Rodriguez et al., 2012) showed AE values in line with the high MODIS-derived AE estimates. Rather than snow cover, the high AE values might be caused by the AE sensitivity to AOD errors, especially in cases where the AOD is very low (Levy et al., 2015). The reference to Rodriguez et al. (2012) has been added in the text.

The cloud-retrieval could be affected by a failure in the cloud mask detecting false clouds instead of snow or ice. The level-3 MODIS atmosphere daily global product daily mean cloud products for each 1° x 1 ° cell are derived from the MODIS cloud mask level-2 product (MYD35_L2). Whether interested in the atmospheric properties of cloud or aerosols, the MODIS Cloud Mask enables the user to quantify the potential errors resulting from cloud contamination by classifying each pixel as either confident clear, probably clear, uncertain, or confidently cloudy trough several spectral test. In general, MODIS cloud detection is based on the principle that clouds' electromagnetic signature makes a scene brighter and colder than what the scene would be if MODIS had a clear view. However, there are situations when the clouds' signature "colder-brighter" is not that clear anymore. One typical situation where often cloud detection is faulty occurs when clouds are located over snow and ice.

*Figure 7: This figure is very nice and informative. Could you please change the colorbar for the COT? The colorbar goes up to 40 but the highest value in the figure is around 20. If you changed this it would be easier to see the trends in the COT.*
The author agrees with the suggestion of the Referee but believes that the original colormap of Fig.7 enables the reader to see the increasing COT as a function of aerosols.

**Response to the technical corrections**
Page 2, line 16: *'in situ' should be changed to 'in-situ'.*
Correction accepted. Text changed accordingly.

Page 3, line 39: *There is no figure 2 b and c.*
The reference was mistakenly addressing Figure 2 instead of Figure 3. The reference has been corrected pointing at Figs. 3b and 3c.

Page 4, line 7: The author changed the verb 'choose' to 'divide'.

Page 4, line 14: *The sentence is somewhat awkward. Please rewrite.*
The sentence is now rephrased as following.
Original: "The difference of these two variables shows which aerosol condition has a larger effect on cloud properties."
Rephrased: "The difference ($\Delta$Cloud_X) between the cloud parameter Cloud_X in clean ($\mathrm{Cloud\_X_{25th\ percentile}}$) and polluted ($\mathrm{Cloud\_X_{75th\ percentile}}$) aerosol conditions evidences the impact on the parameter Cloud_X of these two aerosol cases. "

Page 4, line 20-24: *It seems to me that these sentences presents results and perhaps should be moved to section 4.*
The author agrees with the suggestion and the text in lines 20-24 are moved to the beginning of the Result section.

Page 4, line 43 – Page 5 line 1. *The end of this sentence is confusing since there are no high AOD values over the Atlantic coast of Norway in figure 3a.*
The sentence appear to be missing the adjective 'lowest'. The sentence is now including the adjective: "A decreasing south-north gradient of AOD is observed in Fig. 3a where the highest values are found over Area 3 (Northern-Germany and Poland), and the lowest over Area 2 (the Atlantic coast of Norway and Northern Sweden)."

Page 5, line 9: *"rather unlikely to be correct" awkward, please rephrase.*

The author meant that from previous evaluation studies of the MODIS aerosol product (Levy at al., 2015), a good agreement between AE from MODIS and AERONET stations were found, over water, only in cases for AOD >0.2. This lower limit is not suitable for our area, which has an averaged AOD of about 0.2, therefore the AE's applicability is questionable. Nonetheless, the MODIS AE values are in line with those reported in Melin et al. (2013) over the Baltic Sea and in Rodriguez et al. (2011). The references to Rodriguez et al. (2011) has been added to the text and the references.

The sentence is rephrased as following:

Original: "Over ocean, a good agreement between MODIS AE and AERONET is found globally but with the limitation of AOD > 0.2 (Levy et al., 2015), a restriction that cannot be applied in our study area where the regional AOD is about 0.2. Therefore, the high values of the AE over the Norwegian Sea are rather unlikely to be correct. Nevertheless, the AE over Area 1 (Fig. 3b) is matching the median range of 1.46-1.49 obtained from a validation study that compares the AE retrieved by SeaWiFS and MODIS Aqua/Terra with the three AERONET stations over the Baltic Sea (Melin et al., 2013)."

Rephrased: "Over ocean, a good agreement between MODIS AE and AERONET is found globally with the limitation of AOD > 0.2 (Levy et al., 2015), a restriction that cannot be applied in our study area where the regional AOD is about 0.2. As the sensitivity of AE to AOD errors are especially critical for low AOD values, pixels with AOD <0.2 are expected to have a less qualitatively accurate AE. Nevertheless, the AE over Area 1 (Fig. 3b) is matching the median range of 1.46-1.49 obtained from a validation study that compares the AE retrieved by SeaWiFS and MODIS Aqua/Terra with the three AERONET stations over the Baltic Sea (Melin et al., 2013). Comparable high AE values are collected by Rodriguez et al. (2012) from 2002 to 2011 at the sub-arctic ALOMAR Observatory (Andøya, Norway): the AE peaks during summer season with a multi-annual mean and standard deviation of 1.3 ± 0.4."

Page 5, line 16-17: *The acronym AIE has not been defined. Also the start of the sentence is confusing, the AIE does not say that CER appears to be better correlated with the AOD.*

The definition of the acronym AIE was indeed missing and now it is introduced at Page 2, line 1.

The sentence at Page 5, line 16-17 is rephrased as following:

Original: "According to the first AIE, the CER (Fig. 3e) appears to be better correlated with the AOD (Fig. 3a) rather than the AI (Fig. 3c) and the COT maxima are also in correspondence with the AOD minima over the coast of Norway (Area 2)."

Rephrased: "Considering the theory of the first AIE, that is, an increase in aerosol loading leads to larger CDNC and smaller CER for a fixed LWP, the CER (Fig. 3e) shows correlation with the AOD spatial distribution (Fig. 3a) while worst comparison are found between CER (Fig.3e) and AI (Fig.3c)."

Page 5, line 19: *'which makes these particles very effeicient' please add CCN and change the spelling to efficient.*

Correction accepted. Text changed accordingly.

Page 6, line 4: *'turbulences' change to 'turbulence'.*

Correction accepted. Text changed accordingly.

Page 6, line 10: *There may be other parameters than LTS and aerosol that affects the clouds. I therefore recommend rewriting the first part of this sentence.*

The sentence is rephrased as following:

Original: "To understand to what extent the link between aerosol and cloud parameters are actually due to aerosols, we evaluated the variability of low-level liquid cloud properties as function of aerosol conditions (AOD/AI) and lower troposphere stability (LTS)."

Rephrased: "In an attempt to connect the link between aerosol and cloud with meteorology, we evaluated the variability of low-level liquid cloud properties as function of aerosol conditions (AOD/AI) and lower troposphere stability (LTS)."

Page 6, line 14: *To me it looks like also the CER is affected by the LTS.*
Looking at Fig. 7 b and c, within each AI and AOD bin, the CER changes between 11 and 12 μm in function of LTS. The author consider 1μm to be a rather negligible variation.

Page 6, line 21-25: *This part is confusing to me. There are no results on CF in figure 5 and the structures of the sentences are confusing. Could you please rewrite these sentences to clarify the reasoning with regards to CF results?*
The author agrees with the Referee. The paragraph is, indeed, rather confusing. The text describing the CF results has been modified and rephrased throughout the manuscript according to the discussion presented in the section of the Author's response to General Comments.

Page 7, line 7*: Should ACI be AIE?*
Correction accepted. Text changed accordingly.
The sentence has been rephrased as following:
Original: "This study shows that the different aerosol conditions characterizing the Baltic Sea countries have an impact on the ACI and this can be also observed on a regional scale."
Rephrased: "This study shows that the different aerosol conditions characterizing the Baltic Sea countries contributes to the AIE and this can be also observed on a regional scale."

Page 7*: the conclusion contains quite a bit of discussion of the results. Maybe this section should be renamed Discussion and Conclusion.*
Correction accepted. Text changed accordingly.

**Added Reference**

[revised manuscript text omitted]

---

## Author Comment (AC2) · 31 Jan 2017

**Response to comments from Anonymous Referee #2**

This comment addresses the comments of Anonymous Referee #2. We wish to thank the Referee for the interest in our work and the valuable inputs on the manuscript. The follow document is a point by point response to both the general and specific comments of the Referee.

Note: the following fonts are applied to divide Referee's comments from the Author's response:
*Comments from the Referee*
Response from the Authors
The page and lines numbers refer to the original version of the manuscript. The manuscript following the Author's response is the final revised version.

**General Comments**

*This manuscript estimated the aerosol indirect effect over the Baltic Sea region by using MODIS L3 dataset. Over high latitude regions, such studies are very limited previously because the available dataset are often unreliable. By making use of twelve years of aerosol and cloud properties from MODIS product, the authors investigated the response of the cloud properties to change of aerosol loading based on statistical analysis, and presented some interesting findings over the region. Overall, this manuscript is well written and useful to improve our understanding on aerosol-cloud interaction. The disadvantage is lacking of the detailed explanations and discussions on the results presented (see my specific comments below).*

Each of the specific comments provided by the Referee are addressed below. By discussing the following comments, the Author hopes that the structure of the results and discussion is now better articulated.

**Specific comments**

P2, Line 26-27: *you raised a question here, but we don't see a clear answer finally.*
The Author finds that the paragraph at page 7, lines 5-11 summarizes the answer to the scientific question introduced in the Introduction section.

Page 4, line 7: The Author changed the verb 'choose' to 'divide'.

P4, line 19-24: Fig.2: *Area 3 AOD is much larger than AI, why?*
Looking at Fig.2, the AI values of Area 3 are denoted by the square marker and color coded in red. These values are higher than the AOD.

P5, line 2-3: *"Indicating the dominance of fine particles, high values of the AE are found over the entire Area 1, …', Area 1 should be dominated by sea salt, why the fine particles dominate here?*
The Baltic Sea has a peculiar very low salinity. Therefore sea salt aerosols originated by sea spray are not characteristic of Area 1.

P5, line 18-19: *'Over the Norwegian coast the high values of the COT and the CF can be explained by high hygroscopicity of sea spray aerosols, which makes these particles very efficient'. It seems true, but why we don't see the same thing over the coast of Area 1?*
As stated in the previous comment, the Baltic Sea has a peculiar low salinity. Therefore sea salt aerosols originated by sea spray are not characteristic over the Baltic Sea.

P5, line 25: *should be Fig.4e-h.*
Correction accepted. Text changed accordingly.

P5, line 26-27: *Does MODIS provide cloud top height directly?*

The MODIS cloud top height is provided in the cloud product at L2 but not at L3 (the dataset used in this work).

P5, line 32: *why the CF is not affected by aerosol? Any explanations?*
The author misguided the Referee by stating that no aerosols effect was observed on CF in Fig.3. The Author meant that the signal is not very distinct because the CF lines for the aerosol classes are more 'tangled-up' compared to the profiles of the other cloud parameters.
Figure 3 aims, firstly, to answer the question whether aerosols have an impact on cloud vertical development. Results shows that the highest the aerosols, the lowest is the cloud top pressure (hence higher cloud tops). This effect is observable in each cloud parameter (CF, CER, COT, LWP).The effect of aerosols on CF is not missing from Fig.3, as higher aerosol loading leads to higher vertical development, but this is not a result that is directly linked, and observable, in Fig.6 and Fig.7.
Additionally, Fig. 3 also enables the reader to assess the effect of different aerosol loadings on the cloud parameters. While these are clearly visible for CER, COT, LWP, the signal is not as clear and distinct for CF but is not absent either. The CF for the highest AOD (purple line) is dominantly the highest CF value throughout the vertical profile, in accordance with the AIE's theory. This results is also supported in Fig. 6a and Fig. 7 a,g where high aerosol condition corresponds to higher CF.
The text describing the CF results has been modified as following:
Original: " The opposite behavior, lower average values corresponding to the lower classes of the AI/AOD, can be seen for the COT (Fig. 4c, g) and LWP (Figs. 4d, h) while the CF (Fig.4b, f) is not affected by either the AI or AOD."
Rephrased: "The opposite behavior, lower average values corresponding to the lower classes of the AI/AOD, can be seen for the COT (Fig. 4c, g) and LWP (Figs. 4d, h) while the CF (Fig.4b, f) shows a weaker signal for both AI and AOD cases."

P5, line 40-42: *"The cloud droplet size in Area 1 (the Baltic Sea) and Area 2 (Fennoscandia) shows a strong negative correlation with the AI, while a weak correlation is observed over Area 3 (Central-Eastern Europe)", this is contradictory to our understanding.*
Area 3 shows a contradictory results in respect to the AIEs theories.
The effect of saturation of the cloud response to aerosols might be a reason behind the lower negative correlation between CER and AOD. Supporting this theory we note that for low aerosol loadings (AOD, AI < 0.2), a weak negative slope connect CER to AOD over Area 3.

P5, line 42-43: *'Area 1 has no results for the high LWP bins: clouds over the Baltic Sea are most likely stratiform clouds which are characterized by a lower LWP than for convective continental clouds', any references to present that stratiform clouds hold a lower LWP than convective clouds?*
There is a general relationship between cloud type and LWP as shown by Hess et al. (1998), where it was developed a method for deriving atmospheric radiative properties by modelling aerosols and clouds. The cloud model is created by determining classes of different cloud types and their typical microphysical properties. Marine clouds have fewer droplets than continental clouds of the same type. Nonetheless in smaller number, marine cloud droplets are larger: this results in similar LWP in both environments. Stratus and cumulus clouds, in spite of their very different origin, have about the same LWP. Therefore, the reason behind why the clouds over the Baltic Sea (Area 1) have a lower LWP compared to Area 2 and Area 3 is related to the cloud thickness rather than the cloud type.
The author modifies the sentence as following:
Original: "Area 1 has no results for the high LWP bins: clouds over the Baltic Sea are most likely stratiform clouds which are characterized by a lower LWP than for convective continental clouds"
Rephrased: "Area 1 has no results for the high LWP bins. During summer months, few or no convective clouds form over the Baltic Sea, and mainly thin stratiform clouds are identified in the cloud cover."

P5, line 49-p6, line 1:*'ΔCF (Fig. 6a) presents only positive values suggesting that the CF is always significantly larger in the polluted atmospheric conditions'. ΔCF is always negative as I can see.*
Correction accepted. Text changed accordingly.

P6, line 1-3:*'The positive values of ΔCTP (Fig. 6d) over Area 2 (Fennoscandia) and Area 3 (Central-Eastern Europe) agree with the idea of the vertical development of clouds for higher aerosol loadings (Fig. 4)'. Higher aerosol loadings cause the vertical development of clouds, and then ΔCTP should be negative, correct?*
If higher aerosol loadings enhance clouds vertical development, ΔCTP is positive because cloud top pressure decreased as a function of altitudes. Therefore, from Eq. 2, ΔCTP > 0.

P6, line 6-8*: 'Over land ΔCER is predominantly negative: although small (<2 μm), negative values of the ΔCER indicate that the CER is larger over areas with higher aerosol loadings than over cleaner areas. This result is in contradiction with the theory of the AIEs", is there any explanations for this? From Fig.3, it seems that higher CER correspond to lower aerosol loading, why the contradictory result is shown in Fig. 6?*
Area 3 is the sub-region with overall higher aerosol loadings as we can see from Fig.2 and Fig.3. Figure 3 also shows that there is a connection in the spatial distribution between AOD (Fig.3a) and CER (Fig.3e) but this represents a qualitative results rather than a physical one.
Aerosol conditions (High-AOD and low-AOD cases) and cloud properties are connected in Figure 6. The result showing negative $\Delta CER$ is in contradiction with the first AIE but not necessarily with Fig.2. As we can see from Fig.5, the link between CER and AOD for the Central-Eastern Europe has a weak negative slope, from which we formulated the hypothesis of the saturation of the cloud response to an increase of aerosols.

P6, line 15-16: *'The LWP and CER are negatively correlated with aerosol parameters, showing a stronger response to the AOD than to the AI', CER is negatively correlated with aerosol, but LWP is NOT negatively correlated with aerosol from Fig. 7a.*
The author agrees with the Referee. The LWP is increasing as a function of aerosol loading, with a more distinct signal in the AI case (Fig.7a) than AOD (Fig.7e). The paragraph is modified accordingly.

P6, line 29-30: *'…0.06 to a maximum of 0.16…', what is unit? Please keep consistent with the figure. The ACI values for Area 1 are positive, indicating a positive correlation of CER and aerosol loading, right? But why the correlation coefficients are negative?*
The values there are related to the ACI, a measure per se that is unit less. The ACI as defined in Eq. 3 has a minus sign in front of the formula. Therefore, ACI values are positive and with a negative correlation.

P6, line 37-38: *does this result means that high aerosol loading correspond to larger cloud effective radius for Area 3? Can you give some explanations?*
The relationship between CER and AOD is paradoxically positively correlated over Area 3, meaning high aerosol loading correspond to larger cloud effective radius.
One possible explanation might be the indication of the relationship between CTP and AOD: the CTP decreases for increasing AOD (Fig.4) and at the same time the CER increases with decreasing CTP (higher altitude) in convective clouds (Rosenfeld and Lensky, 1998). Nonetheless, this result must be treated with care as other factors, such as hygroscopic effect, influence the relationship between AOD and cloud parameters and cannot be fully ruled out.
The text above is now included in the manuscript as well as the reference in The Reference section.

**References**

Hess, M., Koepke, P., and Schult, I.: Optical Properties of Aerosols and Clouds: The SoftwarePackage OPAC. Bull. Amer. Meteor. Soc., 79, 831–844, doi: 10.1175/1520-0477(1998)079<0831:OPOAAC>2.0.CO;2, 1998.

[revised manuscript text omitted]